# Sub-Terahertz Imaging-Based Real-Time Non-Destructive Inspection System for Estimating Water Activity and Foreign Matter Depth in Seaweed

**DOI:** 10.3390/s24237599

**Published:** 2024-11-28

**Authors:** Dong-Hoon Kwak, Ho-Won Yun, Jong-Hun Lee, Young-Duk Kim, Doo-Hyun Choi

**Affiliations:** 1Future Technology Foresight Team, Korea Research Institute for Defense Technology Planning and Advancement, Jinju 52852, Republic of Korea; dhkwak@krit.re.kr; 2School of Electronic and Electrical Engineering, Kyungpook National University, Daegu 41566, Republic of Korea; 3Division of Automotive Research, Daegu Gyeongbuk Institute of Science and Technology (DGIST), Daegu 42988, Republic of Korea; howon98@dgist.ac.kr (H.-W.Y.); jhlee@dgist.ac.kr (J.-H.L.)

**Keywords:** terahertz, non-destructive inspection, food quality test, signal processing, object detection

## Abstract

As the importance of hygiene and safety management in food manufacturing has been increasingly emphasized, research on non-destructive and non-contact inspection technologies has become more active. This study proposes a real-time and non-destructive food inspection system with sub-terahertz waves which penetrates non-conducting materials by using a frequency of 0.1 THz. The proposed system detects not only the presence of foreign matter, but also the degree of depth to which it is mixed in foods. In addition, the system estimates water activity levels, which serves as the basis for assessing the freshness of seaweed by analyzing the transmittance of signals within the sub-terahertz image. The system employs YOLOv8n, which is one of the newest lightweight object detection models. This lightweight model utilizes the feature pyramid network (FPN) to effectively detect objects of various sizes while maintaining a fast processing speed and high performance. In particular, to validate the performance in real manufacturing facilities, we implemented a hardware platform, which accurately inspects seaweed products while cooperating with a conveyor device moving at a speed of 45 cm/s. For the validation of the estimation performance against various water activities and the degree of depth of foreign matter, we gathered and annotated a total of 9659 sub-terahertz images and optimized the learning model. The final results show that the precision rate is 0.91, recall rate is 0.95, F1-score is 0.93, and mAP is 0.97, respectively. Overall, the proposed system demonstrates an excellent performance in the detection of foreign matter and in freshness estimation, and can be applied in several applications regarding food safety.

## 1. Introduction

In the modern food manufacturing industry, the importance of hygiene and quality control is being emphasized more than ever. As global food supply chains grow more complex and consumer demands diversify, food manufacturers need to manage quality and safety more rigorously. Consumers are also becoming more focused on food safety and hygiene for their own health, which is putting additional demands on companies. In particular, foreign matters and microbial contamination, which provide significant negative effects on the human body, increase the need for non-destructive and non-contact inspection systems to proactively mitigate these threats. Non-destructive inspection technology allows the assessment of food quality without causing physical damage to the product [1,2,3]. It helps promptly identify and resolve issues that may arise during the manufacturing process. This means that it provides real-time monitoring of food safety without interrupting other manufacturing processes, thereby enhancing operational efficiency and reducing the occurrence of unsafe or defective products.

In food manufacturing processes, one of the representative conventional methods of non-destructive inspection is the vision (image) inspection system [4,5,6,7,8]. Vision inspection systems obtain images of food products using optical devices, such as machine vison cameras, and apply various algorithms to the acquired images in order to detect and sort out defective products or foreign matters. Recently, due to advancements in deep learning, vision inspection systems have demonstrated fast and accurate inspection capabilities. Although this method efficiently uses a color contrast between the product and the foreign matter, it has two significant limitations. First, the vision-based system has difficulty in distinguishing between objects with similar colors and shapes (e.g., black seaweed and black flies). The second problem is that it cannot inspect the internal parts of products. To address these issues, non-destructive inspection methods utilizing various sensors are being actively researched.

To tackle the first problem, hyperspectral imaging inspection [9,10,11,12,13], which performs surface inspections similarly to vision inspection, is an advanced image processing technology that simultaneously collects spectral and spatial information of objects. Since hyperspectral imaging can obtain dozens to hundreds of spectral data across a wide range of wavelengths (e.g., visible and near-infrared (VNIR) [14]), it allows for identification of the chemical composition and physical state of materials. Thus, it is advantageous for distinguishing foreign matters with similar colors to food or detecting changes in the freshness of food. However, it does not solve the second problem by the fact that it still uses light, which cannot penetrate into the food.

To tackle the second problem, the representative methods for inspecting the internal part of products include X-ray inspection and using a metal detector. The X-ray inspection [15,16,17], by using its strong penetrative capability, can be highly effective in detecting hard-typed foreign matters such as metal, bone, stone, etc. However, it struggles to detect soft-typed (low density) foreign matters such as insects, silicone, rubber, etc. Another limitation is that radiation may invoke potential health risks, and it makes it difficult for engineers to work near the equipment. The metal detector [18] is also highly efficient at identifying metallic foreign matters, and it is considered safe by the fact that there is no radiation exposure. However, it can only detect metal objects and cannot detect other hard-typed foreign matters such as plastic and stones. In addition, both devices do not have the function to evaluate the freshness or quality inside the food.

In this study, in order to simultaneously detect foreign matters inside foods and analyze their freshness, we propose a novel sub-terahertz-based inspection system with deep learning algorithms. Terahertz waves occupy a frequency band in the electromagnetic spectrum between microwaves and infrared waves, typically ranging from 0.1 THz to 10 THz. Frequencies below 1 THz are referred to as sub-terahertz waves. These waves have the ability to penetrate and non-destructively analyze the internal structure of food products. Thus, compared to conventional methods, the proposed system has the capability to detect both hard and soft foreign matters inside foods. Additionally, our sub-terahertz-based system [19,20,21,22,23,24] is considered harmless to the human body [25,26], making this method a safer alternative.

In addition to detecting foreign matters, we also propose a method to estimate the depth of foreign matters in food by utilizing the signal variation that occurs with the penetration degree of the foreign matter. In general, terahertz waves do not have penetration capabilities as strong as X-rays. As the material becomes thicker or the distance from the sensor increases, the acquired signals change slightly. Due to this limitation, thick materials cannot be penetrated. However, if the thickness of the food is appropriate, the location and depth of the foreign matter can be inferred by analyzing the signal characteristics of the food and foreign matter.

Regarding the estimation of food freshness, we explored another unique limitation of terahertz waves. Due to the frequency characteristics of terahertz waves, they are highly sensitive to polar substances, particularly water. Consequently, it is challenging to penetrate materials with a high moisture content. Nonetheless, this study used this limitation to its advantage. By analyzing the difference in signal penetration depending on the moisture concentration, we propose a real-time estimation method for water activity levels in dried food. Water activity is an indicator that represents the amount of free water in food and is closely related to the potential for microbial growth and food freshness [27]. When the water activity exceeds the threshold, it can accelerate microbial growth, leading to food spoilage or deterioration. Conventionally, water activity measurement is conducted through destructive methods, where materials are extracted for measurement. This approach is time-consuming and not suitable for application in manufacturing processes. In contrast, the proposed method is non-destructive and provides automatic and rapid estimation of water activity levels. To the best of our knowledge, it is the first attempt to simultaneously explore the real-time inspection for foreign matters with locations and freshness (water activities) in foods. This non-destructive testing system we propose can play an unprecedented and innovative role in food screening based on the characteristics of terahertz waves.

The rest of this paper is organized as follows. Section 2 describes the equipment and materials used in this study. In Section 3, a method for compensating for unstable signals is presented. Section 4 explains the inspection method using sub-terahertz imaging [28,29]. In Section 5, the performance of the inspection model is evaluated and analyzed. Section 6 introduces the inspection platform developed with practical usability in mind and discusses the results of real-site experiments. Finally, Section 7 provides a brief summary and concludes the paper with directions for future research.

## 2. Equipment and Materials

### 2.1. Equipment

In this study, a non-destructive inspection system using sub-terahertz waves at a frequency of 0.1 THz was developed. This system is optimized for the real-time inspection of food products moving rapidly on a conveyor belt. It consists of sub-terahertz equipment (source, scanner), accessories (conical horn antenna, cylindrical concave mirror), a PC, conveyor belt, and mounting frame. The sub-terahertz equipment utilized an impact ionization avalanche transit-time (IMPATT) diode from Terasense Inc., San Jose, CA, USA [30,31]. The detailed configuration of the system is shown in Figure 1 and the specifications of each piece of equipment are presented in Table 1.

For the sub-terahertz source, higher frequencies enhance resolution, and higher power improves penetration capability. We used a high-power source with an output of 800 mW. This sub-terahertz source is equipped with a digital attenuator, allowing the power intensity to be adjusted as needed. For instance, when materials are too thin or have a low density, leading to complete penetration and making them indistinguishable, the digital attenuator can be used to physically reduce the source power to address this issue. When the sub-terahertz source generates waves, the generated waves are focused onto the line-scan-type scanner through a cylindrical concave mirror. For the line-scan-type scanner, there are a total of 256 sensors. Each sensor represents the intensity of signal obtained from the THz source as a 0–1 value. The higher the penetration of the sub-THz signal, the closer the obtained value is to 1. The obtained values are imaged by the inspection platform. The principle of imaging is that the value obtained from each sensor form a pixel, and the pixels obtained from 256 sensors constitute a line. The line-scan-type scanner continuously scans lines to create 2D images. The resulting image has a resolution of 512 × 256 (width × height), and the spatial resolution of each pixel is shown in Table 1. This setup is suitable for monitoring foods on conveyor belts moving at high speed.

The conveyor belt used in this system was constructed from transparent polyurethane (PU). While typical conveyor belts include additives such as fiber or nylon to improve durability and strength, the high-purity transparent PU belt offers superior penetrability for sub-terahertz waves [32,33]. To validate this observation, an experiment was conducted using the same system, where the sub-terahertz signals of seaweed were analyzed under different belts. Sub-terahertz images of seaweed obtained from different belts within the same system are presented in Figure 2. Figure 2a shows the image acquired using a transparent belt, while Figure 2b shows the image acquired using an opaque belt. It can be observed that the sub-terahertz wave penetration in the opaque belt is significantly lower compared to the transparent belt. Transparent PU belts are less strong than belts containing additives, but provide sufficient durability for transporting lightweight materials such as dried seaweed. To further enhance performance, a support structure was added to maintain belt tension and minimize vibrations. This adjustment effectively reduces the potential of signal distortions caused by sagging or belt movement.

The mounting frame serves to securely maintain the distance between the source, mirror, and scanner. Through the repeated calibration and experiments, the optimal distance between core devices was carefully designed: 250 mm from the source to the mirror and 650 mm from the mirror to the scanner, and precisely maintained to enhance the system’s accuracy. This frame allows adjustments to XYZ axes as needed. Such flexibility in distance adjustment ensures optimal signal reception in various environments, thereby maximizing the system’s efficiency.

### 2.2. Materials

#### 2.2.1. Food Selection

In general, sub-terahertz waves are highly sensitive to polar substances, particularly water, due to their frequency band characteristics. This means that terahertz waves are easily absorbed or scattered in materials with a high moisture content, making signal penetration difficult. In addition, even if the moisture content of the sample is sufficiently low, if the thickness exceeds a certain level, it can affect the transmission of sub-terahertz waves, potentially disturbing penetration. Considering these physical limitations, to effectively utilize sub-terahertz waves, it is important to select samples with a low moisture content and an appropriate thickness.

In this study, dried seaweed was selected as a suitable food in terms of moisture content and thickness, considering the physical limitations of sub-terahertz waves. Dried seaweed is a representative export product of Republic of Korea and is consumed worldwide [34]. In particular, a bundle of 100 sheets of dried seaweed is called a “tot” in Korean and is commonly used as a raw material before processing. Dried seaweed contains approximately 5% to 10% moisture, which provides appropriate conditions for sub-terahertz wave penetration. Additionally, the thickness of a tot of seaweed is about 5 cm, making it suitable for sub-terahertz wave penetration. The thickness that sub-THz waves can penetrate varies depending on the system specifications and the material properties. In general, THz penetrates most non-conductive objects, but cannot penetrate metals or dense materials (e.g., stones and bone fragments). In our system, we confirmed that it can penetrate and perform the proposed functions for seaweed up to approximately 7 cm thick. For these reasons, dried seaweed was selected as the target food. Figure 3 shows a tot of dried seaweed.

#### 2.2.2. Foreign Matter Selection

To train the inspection model using an object detection algorithm, sub-terahertz images of foreign matter inside the dried seaweed are required. For this purpose, five representative soft foreign matters were selected for the experiment. The selected foreign matters were silicone, ethylene propylene diene monomer (EPDM), polyvinyl chloride (PVC), polyurethane (PU), and a housefly. These are likely to occur during food manufacturing and processing. The sizes of the foreign matter range from the scanner’s minimum detectable size of 3 mm to 5 mm. Except for the housefly, the shapes of the foreign matter were standardized to a cube. In particular, the reason why the foreign substance sample was made in a cube shape was to normalize the size (width, length, and height) of the foreign matters. This makes it easy to observe how detection performance varies with size. Another reason is that it enhances the performance of the deep neural network by clearly providing ground truth about the size and type of foreign objects during the training procedure. Table 2 shows the foreign matters used in the experiment.

## 3. Signal Compensation

Even when sub-terahertz waves are focused through a mirror, it is difficult to transmit stable signals to all of the scanner’s sensors. This is because various physical factors, such as the imbalance in energy distribution during the focusing process, diffraction, reflection, and scattering of the waves, cause the strength of the signals to be different. To address this non-stability, the signal stability can be improved by applying a signal compensation algorithm.

To compensate for the signal, it is necessary to accurately observe the imbalance in the signal. However, due to the high power of the sub-terahertz source used, all sensors become saturated in the background signal, making it impossible to observe the unstable signal. To address this issue, a digital attenuator can be used to physically reduce the power output. The method for calculating the output when the power is attenuated by x dB can be expressed as follows [35]:(1)$$P_{out}=10^{-\frac{x}{10}}\times P_{in}\;\mathrm{where}\;P_{in}\;\mathrm{is}\;\mathrm{maximum}\;\mathrm{output}\;\mathrm{power}\;\mathrm{of}\;\mathrm{source}\;\left(800\;\mathrm{mW}\right);\;\mathrm{and}\;P_{out}\;\mathrm{is}\;\mathrm{xdB}\;\mathrm{attenuated}\;\mathrm{power}$$

The saturated background signal at maximum output is displayed entirely in white, indicating full saturation (the black lines represent dead pixels). In contrast, when the power was attenuated by 8 dB using the attenuator, the background signal did not result in saturation in all sensors, allowing the observation of signal imbalance. At this time, the attenuated power output was approximately 127 mW.

Signal compensation was applied to resolve the issue of sub-terahertz signals not propagating uniformly across the sensor. This process begins by analyzing the received signal strength for each sensor using the attenuated signal. Then, compensation factors are calculated to adjust the signals to a uniform intensity. Finally, the signal from each sensor is multiplied by a compensation factor. This ensures that the signal intensity received by each sensor is uniformly adjusted to match the level of the sensor with the strongest signal. The compensation factor for each sensor is calculated by dividing each sensor’s intensity value by the maximum value. The method for calculating the compensation factor for each sensor is shown in Equation (2) and the method of applying the compensation factor to the raw signal is shown in Equation (3). By multiplying this pre-obtained compensation factor with the raw signal without power attenuation, the signal can be compensated in real-time.
(2)$$C_{i}=\frac{x_{max}}{x_{i}}\;\mathrm{where}\;C_{i}\;\mathrm{is}\;\mathrm{compensation}\;\mathrm{factor}\;\mathrm{of}\;i_{th}\;\mathrm{sensor};\;x_{i}\;\mathrm{is}\;i_{th}\;\mathrm{attenuated}\;\mathrm{signal}\;\mathrm{and}\;x_{max}\;\mathrm{is}\;\mathrm{maximum}\;\mathrm{value}\;\mathrm{of}\;\mathrm{attenuated}\;\mathrm{signal}$$
(3)$$S_{i}=R_{i}\times C_{i}\;\mathrm{where}\;R_{i}\;\mathrm{is}\;\mathrm{raw}\;\mathrm{signal}\;\mathrm{of}\;i_{th}\;\mathrm{sensor};\;S_{i}\;\mathrm{is}\;i_{th}\;\mathrm{compensated}\;\mathrm{signal}$$

The changes in signal intensity for each sensor before and after compensation can be seen in Figure 4. The results show that, except for the dead pixel at sensor 225, the signal intensity was equalized across all sensors. This method minimized data distortion and improved the reliability and accuracy of the signals.

At the maximum power output, we tested whether the compensation factors pre-obtained through attenuated power remain effective. This can be confirmed through the sub-terahertz images acquired when dried seaweed passes the sensor. Figure 5 presents the dried seaweed sub-terahertz images after compensation. The pre-compensation signals were unstable and exhibited significant noise, shown in Figure 2a. In contrast, the post-compensation signals were stable, with a marked reduction in noise, shown in Figure 5. The surface of dried seaweed is actually not flat, so non-reflective scattering may occur. As seen in Figure 2a, irregular noise patterns caused by non-specular scattering can be observed in areas with weak signals. This noise presents a critical issue by the fact that it could be misidentified as foreign matter (e.g., false positive). This issue was mitigated through the application of signal compensation. This process involves a simple operation of multiplying pre-obtained compensation factors, allowing real-time operation without latency.

## 4. Sub-Terahertz Image-Based Inspection Method

### 4.1. Foreign Matter Depth Estimation

The signals acquired from foreign matters inside the seaweed vary slightly depending on their depth. This is due to differences in refraction and signal attenuation as the distance between the scanner and the foreign matter increases. The experiment was conducted in two cases. In the first case, the samples were classified into three classes: top, middle, and bottom (TMB). The top and bottom classes have a depth of the top 30 sheets and the bottom 30 sheets, respectively, and the middle class has a depth of 30 to 70 sheets. For the second case, the samples were classified into two classes: top and bottom (TB), with each class having a depth of 50 sheets. The thickness of the seaweed and depth range of each class are shown in Figure 6.

We analyzed the data after placing five types of foreign matters at varying depths. The dataset contains foreign matters at different depths within the same class. The results of the analysis show that as the foreign matter is positioned deeper, it moves closer to the scanner, providing clearer shape information. In contrast, when the foreign matter is positioned at shallower depths, it moves farther from the scanner, causing a loss of edge definition and a blurred appearance. Furthermore, it was noted that the color of the foreign matter becomes darker as its depth increases. This trend was observed consistently, regardless of the type or size of the foreign matter. Table 3 shows the signal variations according to the depth at which foreign matters of different types and sizes were positioned.

The estimation of foreign matter depth was performed through the training of an object detection algorithm. The result and performance of this process will be discussed in detail in Section 5.

Estimating the approximate penetration depth of foreign matters provides several benefits. First, areas without foreign matter can be identified, enabling the reuse of intact portions. Second, it shortens the time required to locate the penetrating foreign matter. Lastly, it helps trace the main entry paths of foreign matter, aiding in the implementation of preventive measures.

### 4.2. Water Activity Estimation

Water activity represents the amount of free water in food and plays a crucial role in the freshness and quality of food. When food is not properly dried or external moisture is absorbed during storage and distribution, there is a potential increase in water activity. In particular, high water activity content promotes microbial growth, chemical reactions, and enzyme activity, which leads to food spoilage and deterioration of quality. In general, water activity is measured on a scale from 0 aw to 1 aw, with higher values indicating a greater proportion of free water. For example, microbial growth generally occurs at water activity levels above 0.6 aw, so controlling aw is essential for food preservation and safety. The measurement of aw is achieved using a non-conductive humidity sensor, which uses the equilibrium relative humidity between the material and the air through changes in capacitance, then finally calculates the water activity accordingly.

Currently, there is no method for real-time and non-destructive water activity measurement. Most water activity measurement techniques require the extraction of samples, and take several minutes for each measurement [36]. In this study, a method for real-time estimation of water activity is proposed, utilizing the sensitivity of sub-terahertz waves to moisture.

The optimal dried seaweed has a water activity ranging from approximately 0.5 aw to 0.56 aw, indicating it is safe from microbial growth and can be preserved for an extended period. However, if outside air is absorbed into the seaweed during the drying process or the packaging is damaged during distribution, moisture activity may increase. The sub-terahertz signals observed change incrementally according to the water activity of the dried seaweed. To observe these variations, the water activity was classified into three stages, and the corresponding signal changes were monitored at each level. The water activity was measured using the equipment outlined in Table 4.

Water activity levels were categorized into three stages based on their values: Safe (0.5–0.59 aw), Warning (0.6–0.69 aw), and Danger (0.7 aw and above). The sub-terahertz images of the dried seaweed at different water activity levels indicate that as the water activity increases, the signal intensity decreases, resulting in darker images as shown in Table 5. This characteristic allows for an approximate estimation of the water activity level.

Datasets with different water activity levels were obtained by exposing seaweed to approximately 65% humidity and a temperature of 22 °C for different periods of time. The Safe class includes datasets with water activity levels of 0.5 aw, 0.54 aw, and 0.57 aw; the Caution class includes datasets with water activity levels of 0.62 aw, 0.64 aw, and 0.66 aw; and the Danger class includes datasets with water activity levels of 0.72 aw, 0.8 aw, and 0.85 aw.

### 4.3. Data Collection and Annotation

Training an object detection algorithm requires a large amount of data that includes various cases. The collected data play a crucial role in improving the performance of the trained model. One of the key factors when collecting data is ensuring the balance between classes. If data are excessively skewed toward a particular class, this may negatively affect the training process. In this study, the classes were divided into two cases based on the penetration depth of the foreign matter. The first case is where the foreign matter penetrates the top and bottom, and the second case is where it penetrates the top, middle, and bottom. The dataset was appropriately distributed while considering the balance between classes.

A total of 9659 images were acquired, and the number of objects per class is shown in Figure 7. As a special case, the Safe class contains a relatively larger amount of data due to its frequent occurrence. This helps the model learn enough images of safe states to accurately recognize them. For the remaining classes, an appropriate amount of data was collected to prevent data imbalance, ensuring that the dataset was properly distributed according to the roles (water activity level estimation and foreign matter depth estimation) of each class. The foreign matter classes were abbreviated due to their lengthy names. For example, the “Foreign matter bottom” class was named “FM_Bot”. Although cases of foreign matter contamination or exceeding the threshold for water activity are rare, sufficient data were secured to ensure the model can detect these situations. In order to train the acquired datasets using the object detection algorithm, an annotation process is required. For water activity estimation, bounding boxes are drawn around the entire seaweed area for annotation, while for foreign matters, bounding boxes are placed around the foreign matter area. An example of the annotation process is shown in Figure 8.

For the training of deep learning algorithms, the dataset should be divided into training, validation, and test sets. The training dataset is utilized to optimize the model’s parameters, while the validation dataset is employed to adjust hyperparameters and monitor the model’s performance during training. The test dataset is utilized to evaluate the model’s generalization ability based on untrained data. It is also important to allocate the dataset appropriately into training, validation, and test datasets, considering the total amount of data available. The acquired dataset is relatively small compared to the vast amount typically used for deep learning training, so it was divided into a 60:20:20 ratio, ensuring that sufficient data were allocated to both the validation and test datasets. The proportions of the divided dataset are shown in Figure 9.

### 4.4. Object Detection Algorithm

In this study, YOLOv8 [37] was used as the deep learning algorithm for object detection. YOLOv8 is one of the latest releases in the YOLO series and offers several advantages. First, the model is efficiently designed to maintain high accuracy while achieving very fast processing speeds. Additionally, it utilizes an enhanced feature pyramid network (FPN) structure to extract features at different resolutions. The high-resolution feature maps allow the model to capture fine details, making it highly effective at detecting small objects and handling challenging classification tasks, such as differentiating subtle color contrasts. This capability is advantageous for detecting small foreign matters in food or distinguishing water activity levels based on subtle differences in color contrast. The architecture of YOLOv8 is shown in Figure 10.

YOLOv8 is divided into several sub-models based on complexity, and we utilized the most lightweight version, YOLOv8n. This model reduces computational complexity, allowing for fast processing times even in limited environments, such as without GPU. Since sub-terahertz images are less complex compared to standard RGB images, a high accuracy can still be achieved using the YOLOv8n model. This approach ensures that during system mass production, real-time operation with high accuracy can be maintained while being less dependent on PC performance, resulting in cost savings and other potential benefits.

## 5. Inspection Performance Evaluation

### 5.1. Evaluation Indicators

Precision refers to the ratio of correctly detected objects among the objects detected by the model. In other words, it evaluates how many of the model’s predictions labeled as correct are actually correct. A high precision value indicates that the model has few false detections when identifying objects. The equation is as follows:(4)$$\mathrm{Precision}=\frac{True\;Positive\;\left(TP\right)}{True\;Positive\;\left(TP\right)+False\;Positive\left(FP\right)}$$

Recall refers to the ratio of objects correctly detected by the model out of the total number of actual objects. In other words, it evaluates how successfully the model detects objects from the entire set. A high recall value indicates that the model successfully detects objects without missing any, meaning there are few false negatives. The equation is as follows:(5)$$\mathrm{Recall}=\frac{True\;Positive\;\left(TP\right)}{True\;Positive\;\left(TP\right)+False\;Negative\;\left(FN\right)}$$

The F1-score is the harmonic mean of precision and recall, and it is used to assess the balance between both performance metrics. Since there can be a trade-off between precision and recall, the F1-score provides a single metric that evaluates the overall performance of the model. It helps determine whether precision and recall are well balanced. The F1-score increases when both values are similar, and if one is high while the other is low, the F1-score will be relatively lower. The equation is as follows:(6)$$F1-\mathrm{score}=2\times \frac{Precision\times Recall}{Precision+Recall}$$

Mean average precision (mAP) is a metric used to evaluate the overall performance of object detection across multiple classes. It is calculated by averaging the average precision (AP) for all classes. AP is obtained by calculating the area under the precision–recall curve. The mAP assesses how well the model detects objects across different classes in a balanced manner. A high mAP value indicates that the model consistently detects objects accurately across multiple classes. It is commonly regarded as the most important metric when evaluating a model’s performance. The equation is as follows:(7)$$\mathrm{mAP}=\frac{1}{n}\overset{n}{\underset{i=1}{\sum }}AP_{i}\;\mathrm{where}\;n\;\mathrm{is}\;\mathrm{number}\;\mathrm{of}\;\mathrm{classes}\;\mathrm{and}\;AP_{i}\;\mathrm{is}\;\mathrm{Average}\;\mathrm{Precision}\;\mathrm{of}\;i_{th}\;\mathrm{class}$$

### 5.2. Performance Evaluation Result

To evaluate the performance of the model, datasets from two cases were trained under the same conditions. By training different type datasets under identical conditions and comparing the results, it is possible to assess which type of model performs better. The input data and hyper parameters used in training, and the specifications of the PC used for training are shown in Table 6.

The training results indicate that the TB dataset achieved a higher performance across all evaluation metrics. This is likely due to the FM_Mid class having a higher tendency to be confused with other classes, which may have negatively impacted the overall performance. The performance of the model for each dataset is shown in Table 7.

When evaluating the average precision (AP) for each class, both models exhibited a high performance in the estimation of water activity. In the TMB model, the performance of foreign matter depth estimation for the FM_Bot class was relatively low. This result is attributed to confusion with the FM_Mid class. In contrast, the TB model demonstrated a high performance for both the FM_Top and FM_Bot classes. As a result, the TB model was selected as the model for use. Figure 11 shows the Precision-Recall (PR) curve, and each class’s AP and mAP for both models.

## 6. Sub-Terahertz-Based Real-Time Inspection Platform

To apply the trained model to the real-site processing environment of dried seaweed, a real-time inspection platform was developed. The platform was implemented using the PyQT framework to create a user-friendly graphical user interface (GUI) with an intuitive design. Additionally, the trained YOLOv8n TB model was converted into the Open Neural Network Exchange (ONNX) format [38]. ONNX is a standardized format that enhances model compatibility across various deep learning frameworks, allowing the system to operate seamlessly in different hardware environments. Specifically, ONNX enables stable real-time inspections in CPU environments and on various edge devices. For example, it supports various runtimes and hardware acceleration tools such as TensorRT [39], OpenVINO [40], and ONNX Runtime. This allows our system to run deep learning models with optimized performance on various CPUs, GPUs, and FPGAs.

The platform supports sub-terahertz image monitoring and real-time inspection of dried seaweed moving at high speed on a conveyor belt. The maximum conveyor speed for real-time inspection capability is 45 cm/s. If a foreign matter is detected or the water activity level exceeds the threshold, a buzzer will sound immediately, a red warning screen will flash, and the belt will stop. At this point, the worker can easily identify the issue and remove the defective product. In the future, this system can be upgraded with a rejector function to automatically remove defective products. Examples of the platform’s operation when normal products pass and when defective products are detected are shown in Figure 12.

The platform is operated through a touch screen mounted on the front of the mounting frame. Large buttons and an intuitive control panel were used to enhance the operator convenience, minimizing the risk of operational errors. The platform includes basic functions such as adjusting the scanner settings and emergency stopping of the conveyor belt, as well as additional functions like automatic logging of detected defective products. A usability test conducted in an actual dried seaweed manufacturing facility confirmed that the real-time monitoring and defect detection functions operated smoothly. Figure 13 shows a scene where workers used the platform in an actual dried seaweed manufacturing facility.

## 7. Conclusions

This study proposes a real-time and non-destructive food inspection system using sub-terahertz waves. The system introduces an innovative approach for estimating water activity and the penetration depth of foreign matters in food non-destructively. In general, the terahertz signal can basically penetrate non-conductors, but its penetration performance varies for specific objects according to their density and moisture. The proposed system utilizes the above facts and analyzes the signal intensity of various foreign matters such as insects, silicon, PVC, EPDM, etc. In particular, in order to validate the performance and availability in real manufacturing facilities, we implemented an application platform that cooperates with a 45 cm/s conveyor belt system. For the performance validation of real-time sub-terahertz image analysis, the system utilizes the YOLOv8n model, achieving a high performance in terms of precision, recall, F1-score, and mAP. We believe that the proposed system significantly contributes to enhancing food quality control and safety. This technology is not limited to dried food products, but can also be applied to non-food fields where sub-terahertz wavelengths can penetrate.

In future studies, we plan to analyze the correlation between various biological quality indicators such as mold, microbes, and sub-terahertz images. Additionally, we will also investigate a visual analysis model to predict food quality using deep learning. Furthermore, we plan to explore methods to enhance the low resolution of the terahertz scanner through signal/image processing-based approaches for the detection of smaller foreign matters.

## Figures and Tables

**Figure 1 sensors-24-07599-f001:**
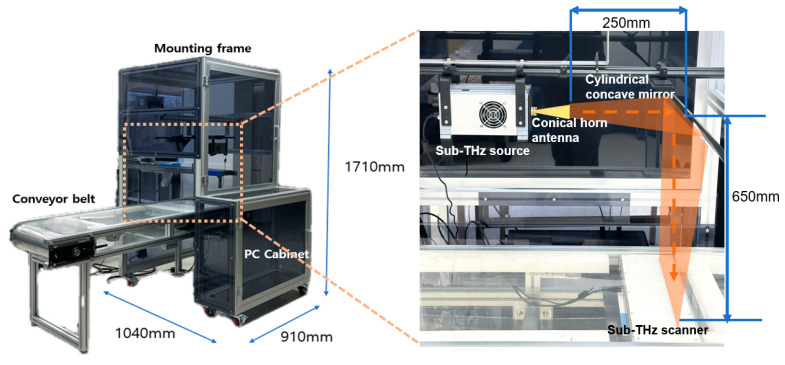
Detailed configuration of the entire system.

**Figure 2 sensors-24-07599-f002:**
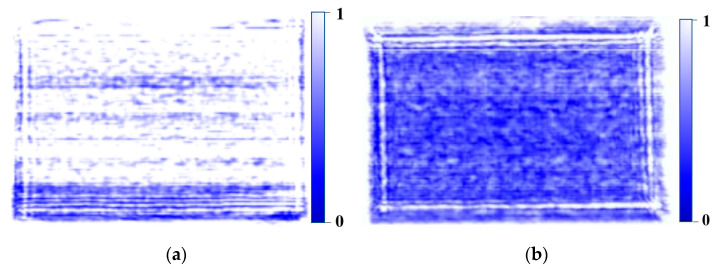
Sub-terahertz seaweed image from different belts: (**a**) transparent belt and (**b**) opaque belt.

**Figure 3 sensors-24-07599-f003:**
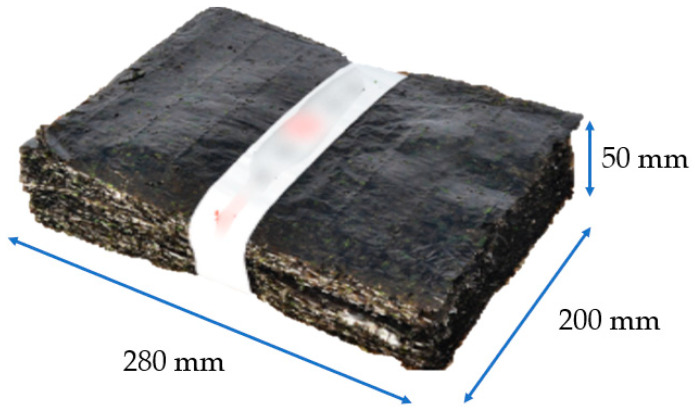
A tot of dried seaweed.

**Figure 4 sensors-24-07599-f004:**
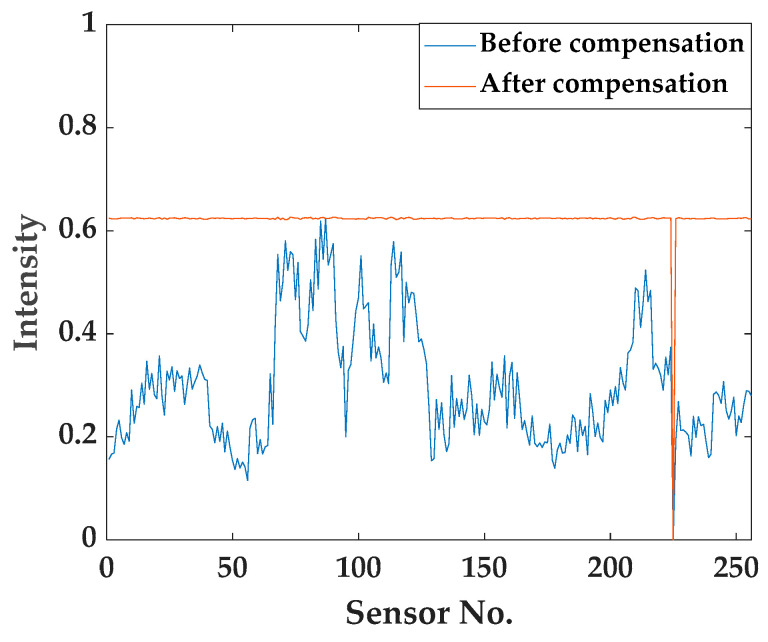
Comparison of signal intensity.

**Figure 5 sensors-24-07599-f005:**
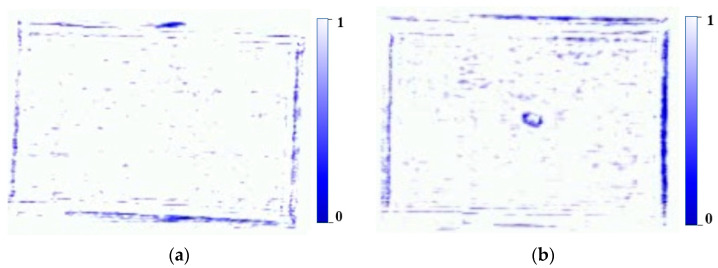
Post-compensated sub-terahertz images: (**a**) seaweed and (**b**) seaweed with foreign matter.

**Figure 6 sensors-24-07599-f006:**
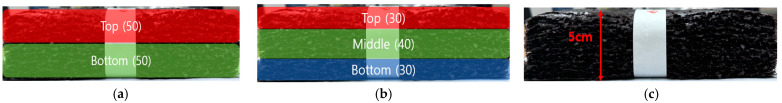
Depth range of each class: (**a**) TB (**b**), TMB, and (**c**) thickness of seaweed.

**Figure 7 sensors-24-07599-f007:**
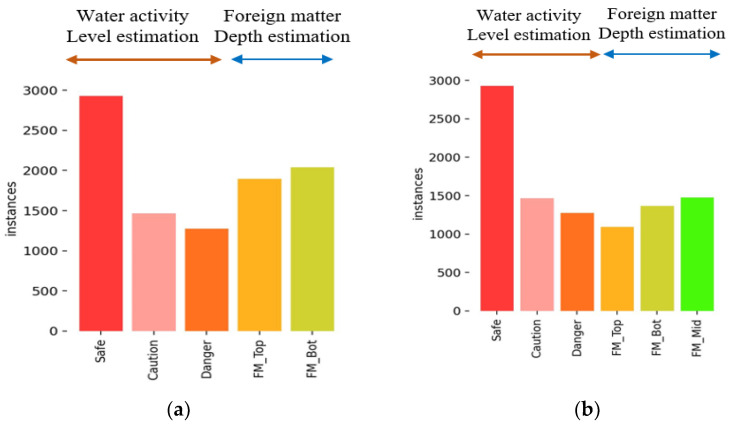
Proportion of acquired data: (**a**) TB Type and (**b**) TMB Type.

**Figure 8 sensors-24-07599-f008:**
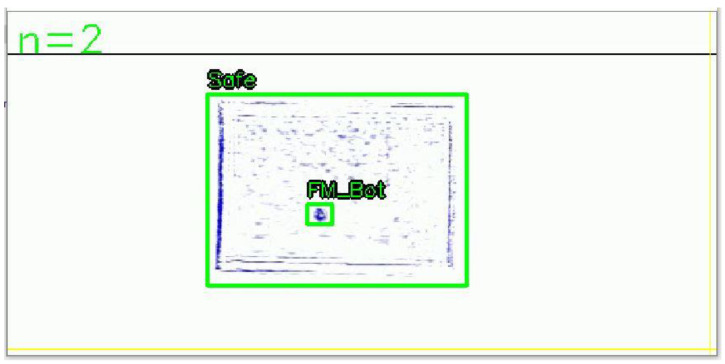
Example of sub-terahertz image annotation.

**Figure 9 sensors-24-07599-f009:**
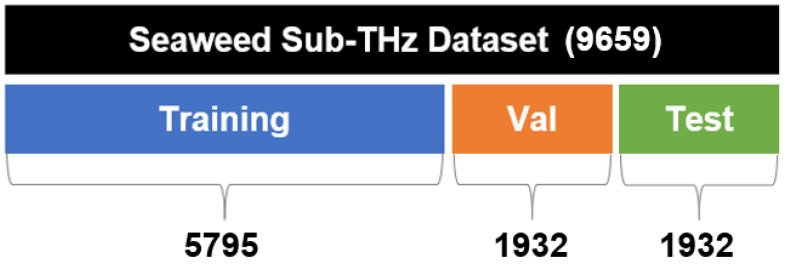
Proportions of the divided dataset.

**Figure 10 sensors-24-07599-f010:**
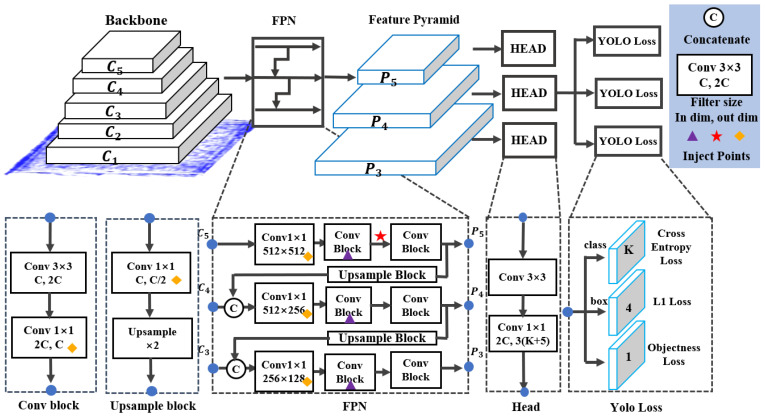
Architecture of YOLOv8.

**Figure 11 sensors-24-07599-f011:**
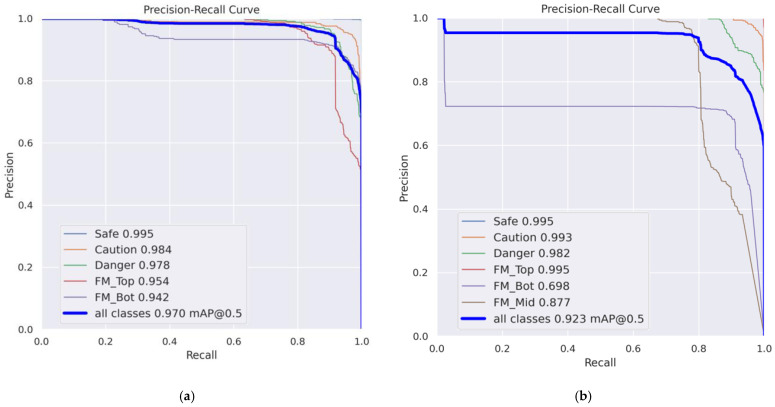
PR curve, each class’s AP and mAP: (**a**) TB model and (**b**) TMB model.

**Figure 12 sensors-24-07599-f012:**
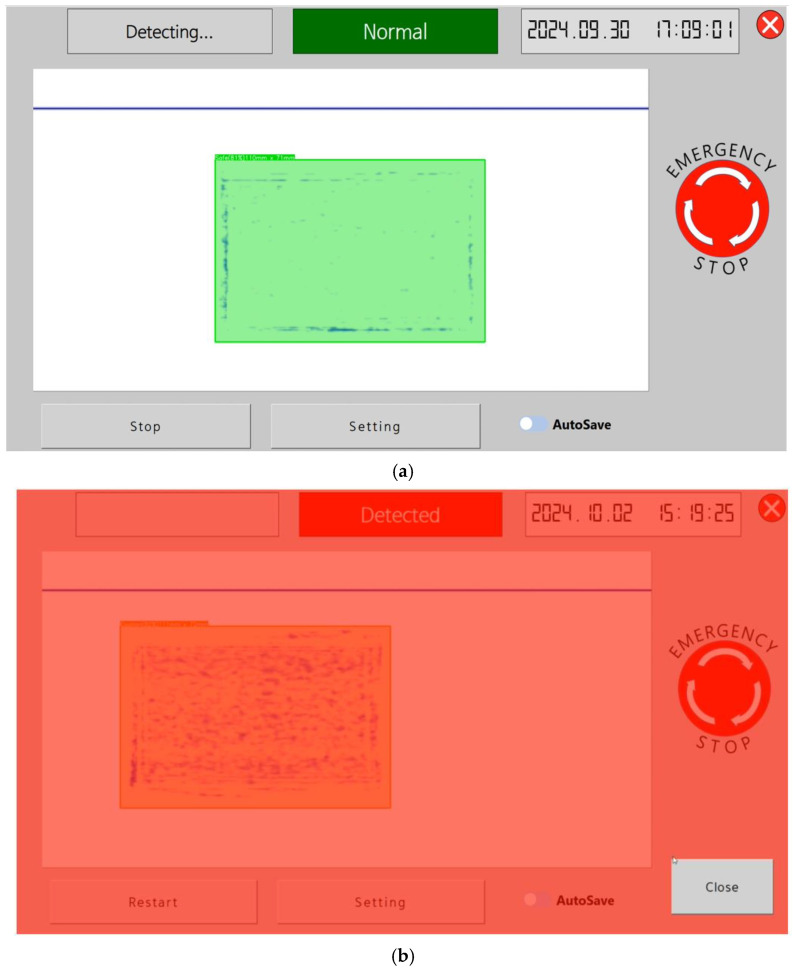
Inspection platform operation example: (**a**) safe product and (**b**) defective product.

**Figure 13 sensors-24-07599-f013:**
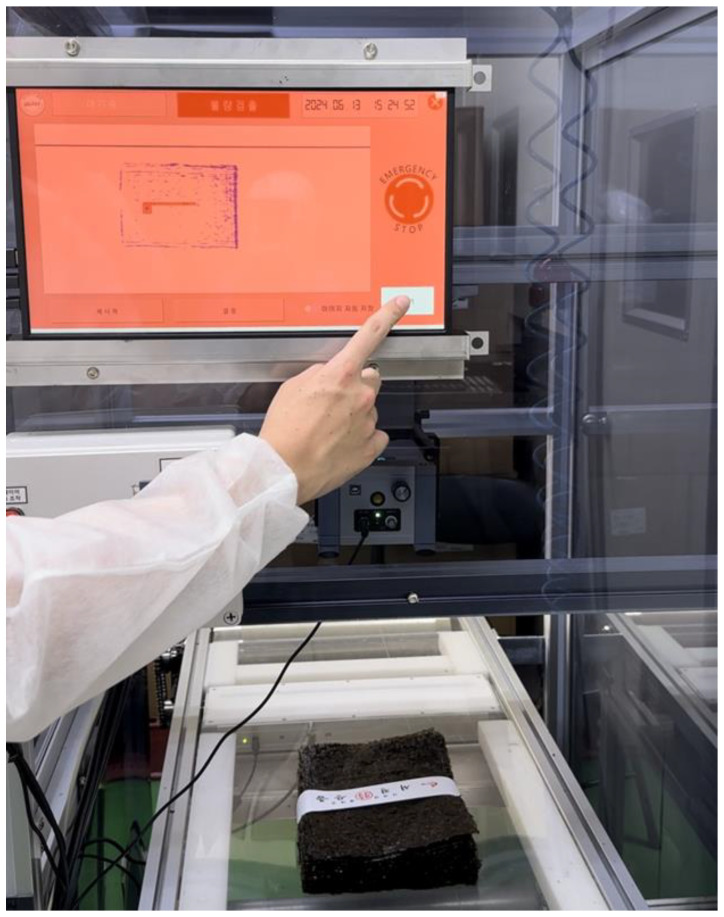
Platform operation scene.

**Table 1 sensors-24-07599-t001:** Specifications of equipment.

Equipment	Index	Specification
Source	Frequency	0.1 THz
	Power	800 mW
Scanner	Number of pixels	256
	Pixel size	1.5 × 3 mm^2^
	Imaging area	384 × 3 mm^2^
Conveyor	Maximum speed	45 cm/s
	Belt	Transparent PU belt
Cylindrical concave mirror	Curvature radius	0.5 m
Conical horn antenna	Aperture size	25 × 6 mm^2^
PC	OS	Windows 10
	CPU	Intel Core i7-10700K 3.80 GHz
	RAM	32 GB
	GPU	GeForce RTX 3060

**Table 2 sensors-24-07599-t002:** Foreign matters used in the experiment.

Type	Silicone	EPDM	PVC	PU	House Fly
Image	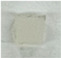	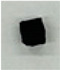	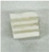	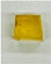	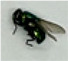
Size [mm]	3–5	3–5	3–5	3–5	4.5 × 8 × 3.5(W × H × T)

**Table 3 sensors-24-07599-t003:** Foreign matter signal changes according to depth.

Foreign Matter	Size	Top	Middle	Bottom
PVC	3 mm	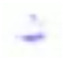	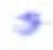	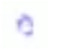
5 mm	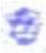	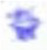	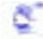
Silicon	3 mm	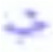	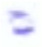	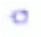
5 mm	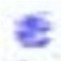	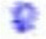	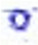
EPDM	3 mm	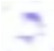	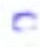	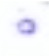
5 mm	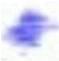	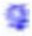	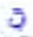
Entry 4	3 mm	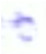	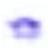	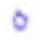
5 mm	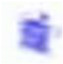	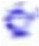	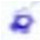
House fly	5 mm × 8 mm	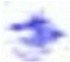	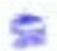	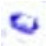

**Table 4 sensors-24-07599-t004:** Water activity measurement equipment image and detailed specifications.

Image	Index	Specification
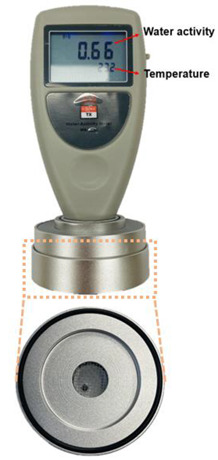	Measuring range	0–1.0 aw
	Resolution	0.01 aw
	Accuracy	±0.03 aw
	Sensor type	Non-conductive humidity sensor
	Sampling time	5 min.
	Size	135 × 70 × 44 mm

**Table 5 sensors-24-07599-t005:** Sub-terahertz images for each level of water activity.

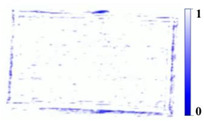	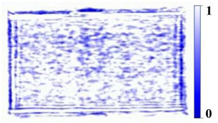	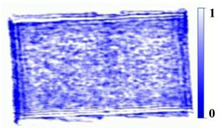
Safe (0.5–0.59 aw)	Caution (0.6–0.69 aw)	Danger (0.7 aw–)

**Table 6 sensors-24-07599-t006:** Training information and PC specifications used for YOLOv8n training.

Type	Index	Information
Input	Image size	640 × 640
	Amount	9659
Hyper parameters	Epochs	100
	Batch size	32
	Learning rate	0.01
PC	CPU	Intel i9 13900K
	GPU	Geforce RTX 4090 D6X 24 GB × 2
	RAM	128 GB

**Table 7 sensors-24-07599-t007:** Model performance for each dataset.

Dataset Type	Precision	Recall	F1-Score	mAP
TB	0.91	0.95	0.93	0.97
TMB	0.87	0.92	0.89	0.92

## Data Availability

The datasets presented in this article are not readily available due to manufacturers’ confidentiality policy.

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
