# Peer review of "Sub-Terahertz Imaging-Based Real-Time Non-Destructive Inspection System for Estimating Water Activity and Foreign Matter Depth in Seaweed"

_sensors, 2024, doi:10.3390/s24237599_

Round 1
Reviewer 1 Report
Comments and Suggestions for Authors
The article " Sub-Terahertz Imaging Based Real-Time Non-Destructive Inspection System for Estimating Water Activity and Foreign Matter Depth in Seaweed " form Dong-Hoon Kwak et al., presents a real-time and non-destructive food inspection system using sub-terahertz waves.
The content of the article is interesting. It shows how they can simultaneal estimate water activity through different microorganisms for assessing the freshness of seaweed by analysing the transmittance of signals within the THz image, and the penetration depth of different elements in food such as silicone, Ethylene Propylene Diene Monomer (EPDM), Polyvinyl Chloride (PVC), Polyurethane (PU),) in a non-destructively way (by utilizing the signal variation that occurs with the penetration degree of the foreign matter). The system employs the deep learning model (YOLOv8n) for embedded devices.
Moreover, the topic is suitable for a journal like Sensors, since it addresses the food safety topic with a sensor. However, there are some things that need to be addressed.

English language and grammar need amendment in terms of the sentence structures and the tense used; suggest doubling check tenses for the whole text.
Reviewer 2 Report
Comments and Suggestions for Authors
Article is very interesting and great results and applications of THz spectroscopy. Please consider to address my comments to strengthen it

Reviewer 3 Report
Comments and Suggestions for Authors
Although the overall topic of the manuscript " Sub-Terahertz Imaging Based Real-Time Non-Destructive Inspection System for Estimating Water Activity and Foreign Matter Depth in Seaweed” is relevant and interesting, I have listed some issues that need to be clarified by the author when evaluating the manuscript. In my opinion, some improvements are essential prior publishing in scientific reports. My recommendation is “Major revision”.
1). What’s the roughness of dry seaweed studied? Non-specular scattering should be considered if the sample is not uniform. This information should be mentioned in the manuscript.
2). The THz detector should be added to the inspection system.
3). Figure 4 (a) is missing.
4). What is the lateral resolution of THz image could be achieved?
5). How long it will spend to obtain Figure 6
6). It is impossible to distinguish the exact location as well as the geometry of foreign objects hidden in the dried seaweed, As shown in table 2, why not use some signal/image processing approaches to improve the depth and lateral resolution? (what’s the maximum penetration depth the inspection system can detect)
7). Table 3 is meaningless to this paper, Please delete it. (It only present the growth potential of different types of microorgansims under various aw range. )
8). The equipment in table 4 seems to be not suitable for large area, while the water activity is is position-dependent.
9). Colorbar should be added to the THz image in Table 5.
10). As shown In Fig. 13, the dried sea weed is packed with paper belt, dies the paper belt will be removed prior to THz measurement?
11). From my perspective, the curved front surface of dried seaweed has negative impact to the intensity of collected THz signals, which will lower the accuracy of water content estimation as well as foreign object identification only based on the amplitude of THz signals.
12). It is necessary to add the reference:
“Zhai M, Locquet A, Citrin DS. Terahertz imaging for paper handling of legacy documents. Sensors. 2021 Oct 12;21(20):6756.”
13). What’s the distance between the cylindrical cocave mirror and the surface of coveyor belt?
14). How to achieve a good focus on the surface of dried seaweed? In Figure 13, the spot of focus will not on dried seaweed if the dried seaweed studied is too thick/thin.
Comments on the Quality of English LanguageThere are several errors in English grammar. Please check carefully.
Round 2
Reviewer 1 Report
Comments and Suggestions for Authors
Thank you for the revisioned version. It is now clearer, more accurate and in-depth
Reviewer 2 Report
Comments and Suggestions for Authors
Hello,
Great work and great improvement.
I would suggest to integrate one more citation: Terjani, Badr, et al. "Plastic particles detection in animal flour using continuous terahertz waves." 2023 Photonics North (PN). IEEE, 2023.
Also, please double check last time little typos, such as space between figure number and letter etc.
Otherwise, it looks great. Thank you for addressing all my comments! Happy for THz improvement and more industrial applications!
Reviewer 3 Report
Comments and Suggestions for Authors
No further comments